# Effects of water stably-enriched with oxygen as a novel method of tissue oxygenation on mitochondrial function, and as adjuvant therapy for type 2 diabetes in a randomized placebo-controlled trial

**Joan Khoo**[1]*, **Christoph E. Hagemeyer**[2], **Darren C. Henstridge**[3,4], **Sumukh Kumble**[2], **Ting-Yi Wang**[2], **Rong Xu**[2], **Linsey Gani**[1], **Thomas King**[1], **Shui-Boon Soh**[1], **Troy Puar**[1], **Vanessa Au**[1], **Eberta Tan**[1], **Tunn-Lin Tay**[1], **Carmen Kam**[5], **Eng-Kiong Teo**[1]

1 Department of Medicine, Changi General Hospital, Singapore, Singapore, 2 NanoBiotechnology Laboratory, Monash University, Melbourne, Victoria, Australia, 3 Baker Heart and Diabetes Institute, Melbourne, Victoria, Australia, 4 School of Health Sciences, University of Tasmania, Launceston, Tasmania, Australia, 5 Clinical Trials and Research Unit, Changi General Hospital, Singapore, Singapore

* joan.khoo.j.c@singhealth.com.sg

## Abstract

### Background

Diabetes mellitus is associated with inadequate delivery of oxygen to tissues. Cellular hypoxia is associated with mitochondrial dysfunction which increases oxidative stress and hyperglycaemia. Hyperbaric oxygenation therapy, which was shown to improve insulin sensitivity, is impractical for regular use. We evaluated the effects of water which is stably-enriched with oxygen (ELO water) to increase arterial blood oxygen levels, on mitochondrial function in the presence of normal- or high-glucose environments, and as glucose-lowering therapy in humans.

### Methods

We compared arterial blood oxygen levels in Sprague-Dawley rats after 7 days of ad libitum ELO or tap water consumption. Mitochondrial stress testing, and flow cytometry analysis of mitochondrial mass and membrane potential, were performed on human HepG2 cells cultured in four Dulbecco's Modified Eagle Medium media, made with ELO water or regular (control) water, at normal (5.5 mM) or high (25 mM) glucose concentrations. We also randomized 150 adults with type 2 diabetes (mean age 53 years, glycated haemoglobin HbA1c 8.9% [74 mmol/mol], average duration of diabetes 12 years) to drink 1.5 litres daily of bottled ELO water or drinking water.

### Results

ELO water raised arterial oxygen tension pO2 significantly ($335 \pm 26$ vs. $188 \pm 18$ mmHg, $p = 0.006$) compared with tap water. In cells cultured in control water, mitochondrial mass and

**Data Availability Statement:** All relevant data are within the paper and its Supporting information files.

**Funding:** The study was sponsored by Elomart Pte Ltd, Singapore. As our study was sponsored by Elomart Pte Ltd, we are including this statement in the cover letter that Elomart Pte Ltd had no role in study design, data collection and analysis, decision to publish, or preparation of the manuscript. The authors have no competing interests relating to employment, consultancy, patents, products in development or marketed products relevant to this study.

**Competing interests:** Sponsorship of this study by Elomart Pte Ltd does not alter the authors' adherence to PLOS ONE policies on sharing data and materials.

membrane potential were both significantly lower at 25 mM glucose compared with 5.5 mM glucose; in contrast, mitochondrial mass and membrane potential did not differ significantly at normal or high glucose concentrations in cells cultured in ELO water. The high-glucose environment induced a greater mitochondrial proton leak in cells cultured in ELO water compared to cells cultured in control medium at similar glucose concentration. In type 2 diabetic adults, HbA1c decreased significantly (p = 0.002) by 0.3 ± 0.7% (4 ± 8 mmol/mol), with ELO water after 12 weeks of treatment but was unchanged with placebo.

## Conclusions

ELO water raises arterial blood oxygen levels, appears to have a protective effect on hyperglycaemia-induced reduction in mitochondrial mass and mitochondrial dysfunction, and may be effective adjuvant therapy for type 2 diabetes.

## Introduction

Oxygen is essential for maintenance of life and normal metabolic function in humans but is often overlooked in the management of diseases related to disturbances of metabolism [1]. Numerous disease processes are associated with cellular hypoxia, in particular diabetes mellitus, for which postulated mechanisms include impaired release of oxygen from haemoglobin [2], slowed haemoglobin oxygen saturation [3], defective mitochondrial oxidative phosphorylation [4] and impaired insulin signalling [5]. Hyperglycaemia also exacerbates hypoxia [6] which worsens insulin resistance [7], induces pancreatic beta-cell dysfunction through oxidative stress [8, 9], and beta-cell de-differentiation which decreases functional beta-cell mass [10], resulting in a vicious circle of hypoxia and hyperglycaemia.

Hyperbaric oxygen therapy (HBOT) increases the amount of oxygen dissolved in plasma [11], thereby facilitating diffusion of dissolved oxygen directly into tissue fluid and cellular water. HBOT was shown to improve tissue oxygenation and insulin sensitivity in hyperglycaemic overweight and obese men [12] but is not practical for long term management of diabetes because it is expensive, inconvenient to administer and carries risks of barotrauma. Hydration with plain water per se has been shown to have benefits for glycaemic control [13–15]. Conversely, dehydration aggravates hyperglycaemia [16] with consequent worsening of glycaemic control, demonstrating that adequate water intake contributes to lowering glucose levels in diabetes. Oxygen in ingested water can be absorbed into the circulation from the gastrointestinal tract through aquaporins in the intestinal epithelium [17], which makes oral delivery of dissolved oxygen viable for oxygen supplementation.

The challenge of using water as a vehicle for oxygen delivery is the naturally low solubility of oxygen in water in the absence of hyperbaric pressure [11]. If oxygen can be dissolved in water at higher than usual concentrations and its concentration of dissolved oxygen stably maintained in solution in a normobaric environment, it would be possible to use drinking water to deliver oxygen. We identified a commercially available oxygen-enriched bottled drinking water (ELO water) that could stably maintain a higher dissolved oxygen concentration and utilised it to test our hypothesis that gastrointestinal absorption of oxygen in drinking water could raise arterial oxygen level. We also investigated the effects of increased dissolved oxygen content, using ELO water, on mitochondrial mass and function in cultured HepG2 cells in the presence of a high-glucose medium, postulating that the deleterious effects on

mitochondria of an environment which mimics hyperglycaemia in diabetes would be mitigated by ELO water but not normal (control) water. As HBOT improved insulin sensitivity and glycaemic control through increased tissue oxygenation [12], we hypothesized that the improved tissue oxygenation provided by ELO water would lower blood glucose in a setting of diabetes. Our overall aim was to evaluate ELO water as an adjuvant treatment for diabetes, which was achieved by comparing the effects of drinking ELO water on glycaemic control in adults with type 2 diabetes in a randomized trial with placebo (normal) drinking water as the control.

## Materials and methods

### Testing ELO water for dissolved oxygen stability

A variety of water oxygenation techniques, such as high-pressure injection of oxygen, addition of ozone, use of ultrasound and electrolysis, have been used to produce commercially-available oxygen-enriched water, with varying degrees of stability of dissolved oxygen. While we have no access to the trade secret proprietary method used to increase oxygen solvation in ELO water, we have been informed by the manufacturer (Elomart Pte Ltd, Singapore) that their technology does not utilise high pressure nor ozonation. ELO water can be obtained directly for research purposes by contacting Elomart Pte Ltd through their corporate email addresses (contact_au@elomart.com, contact_sg@elomart.com).

We tested the oxygen stability of ELO water by opening a 1.5 L bottle of ELO water and inserting the probe of a dissolved oxygen meter (HI2040, Hanna Instruments) into the bottled water every 24 hours to obtain a reading of the dissolved oxygen (DO). During the period of 7 days, the bottle of ELO water was left open to the atmosphere. To test the stability and longevity of DO in ELO water with the bottle opened and recapped, the DO level was measured every 6 to 7 days over a period of 58 days. The Hanna DO meter works on the Clark's electrode principle of dissolved oxygen measurement.

### Testing effects of ELO water on arterial oxygen in animal models

Eight male six-week-old Sprague-Dawley rats (AMREP Animal Services, Melbourne) were housed two to a cage and kept under standardised environmental conditions (20±1˚C, 50 ±10% relative humidity, 12 hrs light-dark cycle) and received food, water and bedding. The animals were divided into two cohorts of 4 rats each, which drank either ELO water or control (tap) water. Each cage, containing two animals, was provided with 100 ml of either ELO or control water, so that each animal could consume a standard volume of 40 ml per day. Water supplies were accessible ad-libitum and exchanged with fresh supplies every 24 hours for seven days. The group size was calculated based on expected effect size and statistical power intended (significance level of 0.05 and a power of 0.8). Work was conducted in accordance with the Australian code for the care and use of animals for scientific purposes, wherein animal numbers are kept to the lowest possible while achieving the required statistical power for the study (https://www.nhmrc.gov.au/about-us/publications/australian-code-care-and-use-animals-scientific-purposes).

All 8 rats were sacrificed on the same day. Each rat was anesthetised with isoflurane (5%) and 100% oxygen was given through a snout cone given at flow rate of 1L/minute, and to each animal throughout the procedure using a well-fitting snout cone. The same snout cone and oxygen delivery rate of 1 L/minute was used for all the animals. Inhaled oxygen supplementation during isoflurane anaesthesia is a strict requirement of the Alfred Medical Research and Education Precinct Animal Ethics Committee (Melbourne, Australia) where these studies were performed. With this method of 100% oxygen delivery through a tight-fitting snout cone,

rats were expected to register an arterial pO2 that is higher than breathing room air [18]. Hence, any effect of ELO water on the arterial pO2 has to offset the rise in oxygen contributed by the inhaled oxygen supplementation during anaesthesia. Isoflurane (Henry Schein) was dropped to 2% to maintain a stable level of anaesthesia, which was monitored through reflex response and breathing frequency. The abdomen was wet and cleaned with 70% ethanol. A xipho-pubic skin incision was made with a blade and abdominal wall was opened in layers. The posterior peritoneum was carefully dissected to expose the abdominal aorta. Each procedure was completed within the same day by the same operator. The order in which the animals from both cohorts had the surgery was randomised. The blood samples were analysed by an independent technician at the Alfred Hospital who was not part of the study and who was blinded to the study treatment arms.

Dedicated heparin coated syringes attached to 25-G needle supplied by the Alfred Hospital Pathology Laboratory were used for blood sampling from the aorta and these were used to obtain 1 ml of arterial blood from the abdominal aorta for blood gas testing. All blood samples in syringes were expelled of air bubbles and rocked from side to side to achieve homogenous heparinisation, syringes were placed on ice and analysed in the Alfred Hospital Pathology Laboratory using a Siemens Rapid point 500 within 30 minutes. The rats were then killed by cardiac incision and cervical dislocation while under anaesthesia. Death was confirmed by absence of cardiac pulse before disposing of the carcass. The animals' care and all experimental procedures were carried out in strict accordance with the recommendations in the Guide for the Care and Use of Laboratory Animals of the National Health and Medical Research Council, and approved by The Alfred Medical Research and Education Precinct Animal Ethics Committee (E/1767/2017/M).

## Mitochondrial stress tests in vitro

A mitochondrial stress test (Agilent Technologies, Catalogue #103015–100) was conducted in HepG2 cell cultures using the Xfe96 extracellular flux Analyser which uses a plate-based approach and fluorescence detectors to accurately and simultaneously measure cellular oxygen consumption rates (OCR) of multiple samples in real time [19]. HepG2 cells were cultured in media of different glucose concentrations made with either ELO or control water over a 2-week period. Powdered Dulbecco's Modified Eagle Medium (DMEM) media was reconstituted in ELO water or normal water and the pH adjusted to a level between 7.2–7.4. The media were 1) normal (5.5 mM) glucose concentration-control water DMEM, 2) normal (5.5 mM) glucose concentration-ELO water DMEM, 3) high (25 mM) glucose concentration-control water DMEM or 4) high (25 mM) glucose concentration-ELO water DMEM. Each type of medium was freshly made up for each passage. During the two-week passaging of cells, each time the media was replaced, the freshly made ELO water medium ensured that the ELO cells were always bathed in more dissolved oxygen than the control cells with control medium. In order to determine if high levels of dissolved oxygen were retained following reconstitution of DMEM powder in ELO water, an Edge® oxygen meter probe (Hanna Instruments Australia) was immersed into the media where higher levels of dissolved oxygen were detected at 14 ppm, relative to control water, which had dissolved oxygen at 7 ppm.

A cell count was conducted and 10,000 cells from each treatment group were seeded per well in a Seahorse bioanalyzer 96-well plate. After 24 hours of incubation the HepG2 cells were assayed for mitochondrial function by following manufacturer's (Seahorse Bioscience) protocols. Oligomycin (1.5 μM), FCCP (1 μM) and rotenone/antimycin A (0.5 μM) were sequentially injected and OCR recorded. The Seahorse assays were analysed using Wave software, according to the manufacturer's instructions. OCR measurements were used to calculate the

activity associated with basal respiration, oxygen consumption linked to ATP production, and the proton leak across the inner mitochondrial membrane, following established methods [20]. The minimum OCR after rotenone and antimycin A injection was interpreted as the OCR due to non-mitochondrial respiration, and this rate was subtracted from all other measurements.

## Flow cytometry analysis of mitochondrial mass and membrane potential

Mitochondrial mass was assessed in HepG2 cells by labelling with MitoTracker Green (Thermo Fisher M7514), which selectively stains all undamaged mitochondria regardless of the membrane potential. The changes in mitochondrial membrane potential (reduction suggesting mitochondrial dysfunction) were assessed by labelling with tetramethylrhodamine ethyl ester (TMRE) (Abcam, ab113852) [21].

In brief, HepG2 cells were cultured in freshly made DMEM media (pH 7.4) in different glucose concentrations with DMEM made either with ELO water or control water in each passage over 14 days period. Those media conditions were 1) normal (5.5 mM) glucose concentration-control water DMEM, 2) normal (5.5 mM) glucose concentration-ELO water DMEM, 3) high (25 mM) glucose concentration-control water DMEM or 4) high (25 mM) glucose concentration-ELO water DMEM. The freshly made ELO water medium ensures sufficient dissolved oxygen (DO) in ELO treated cells compared with control water over two-week culture. The DO of ELO medium is around 14 ppm, while control water is around 7 ppm measured using an Edge® oxygen meter probe (Hanna Instruments Australia). HepG2 Cells (1 x10^5 cells in 200 μL medium, unfixed) from each treatment group were stained with 100 nM MitoTracker Green and 100 nM TMRE for 15 minutes at 37 degrees Celsius, and washed with FACS buffer (2% FBS in PBS) and centrifuged at 500 g, 5 min, and resuspended in 200 ul FACS buffer, and 10,000 cells were used in each treatment group for analysis on a FACS CantoII (Becton, Dickinson & Company) using FlowJo v10.7 software (BD).

## Testing effects on glycaemic control of adults with type 2 diabetes

We conducted a randomized, double-blind, placebo-controlled, proof-of-concept study of ELO water in adults with type 2 diabetes in Singapore recruited between March 2017 and July 2018. The protocol and amendments were approved by the Singapore Health Services ethics committee and institutional review board (IRB 2016/2606). The study was implemented according to Good Clinical Practice and the Declaration of Helsinki, and registered at clinictrials.gov (identifier: NCT04127890). Due to the trial sponsor's (Elomart Pte Ltd) concerns that intellectual property could be plagiarized if details of the study were publicly available, the study was registered after enrolment of participants was complete. The authors confirm that all ongoing and related trials for the use of ELO water for glycaemic control in type 2 diabetes are registered. We recruited men and women aged 21–70 years with type 2 diabetes and haemoglobin A1c (HbA1c) at screening of 8.0–11.0% (64–97 mmol/mol) on oral glucose-lowering medication and/or insulin at a stable dose for the last three months on follow-up in specialist diabetes outpatient clinics in Changi General Hospital (CGH), or with advertisement posters on the premises of CGH. Subjects were excluded if they had previously drunk ELO water, had undergone bariatric surgery, were on weight-loss medication, or had eGFR < 30 ml/min, haemolytic anaemia, hemoglobinopathies or comorbidities necessitating fluid restriction. All subjects gave written informed consent prior to screening.

Taking the medium effect size (Cohen's d = 0.5) between ELO arm and placebo to detect a difference in reduction of HbA1c by 0.5%, we recruited 75 subjects per arm (total 150) for the study to have 80% power with significance level of 5%, accounting for dropout rate 15%.

Subjects were stratified according to their sex and randomized by an independent biostatistician using a computer program in a 1:1 ratio to two groups coded as A or B, to receive either ELO drinking water or bottled drinking water supplied from the same sponsor (Elomart Pte Ltd, Singapore) (Fig 1). The group assignments were concealed in sequentially-numbered opaque sealed envelopes. Investigators, subjects and study statisticians were all blinded to the code. An independent biostatistician who was not part of the study team was responsible for preparing the randomization numbers in individual sealed envelopes. Only one research co-ordinator was holding the envelopes. Each sealed envelope was opened in numerical order, in the presence of each subject only after he or she had signed the informed consent form, and the randomization number was then assigned accordingly. The dates of randomization were also recorded at the same time.

The water was delivered to each subject's home at regular intervals by an independent logistics team, in bottles that were indistinguishable except for labels (A or B), with delivery co-ordinated by a CGH staff member not involved in the study. Subjects were provided with a diary to record their consumption of the water, which was checked by the research co-ordinator at 2-weekly visits for 12 weeks. Minimum compliance was defined to be at least 85% of the water (12/14 bottles/fortnight), equivalent to 9 litres/week. They were advised not to change their pre-study diet or exercise habits for the study duration, and their healthcare providers were given written advice to avoid changing medications for the duration of the study.

## Statistical analysis

Statistical analyses were performed with SPSS version 21.0. The primary efficacy end point was the change in HbA1c at 12 weeks. This had been amended from 24 weeks in the original protocol based on findings of a study in women with type 2 diabetes published after the start of our recruitment which showed that increased water intake lowered glucose and HbA1c level within 12 weeks [15].

The two-tailed Students t-test on two independent variables, assuming an unknown variance and a p-value $< 0.05$, was used for the animal and cell studies. For the study of diabetic subjects, a general linear model with repeated measures and a one-sided significance level ($P < 0.05$) was used to evaluate within- and between-group differences from baseline to 6 and 12 weeks. Subjects with missing data were excluded from the analysis by default. We conducted sensitivity analyses using a last observation carried forward strategy as planned *a priori*, of which the results were similar and hence not presented.

## Results

### Stability of dissolved oxygen in ELO water

The average dissolved-oxygen (DO) level in ELO water (Fig 2A) was significantly ($p < 0.001$) higher ($16.40 \pm 1.11$ ppm), compared to tap water ($8.31 \pm 0.38$ ppm). The DO level in ELO water remained high for 58 days despite daily opening and recapping of the bottle (Fig 2B).

### Effects of ELO water vs. tap water on arterial oxygen level in rats

The rats which drank ELO water achieved a mean arterial pO2 of $335 \pm 26$ mmHg, which was significantly ($p = 0.006$) higher than that of the control group (Fig 3). The control rat group mean arterial pO2 was raised at $188 \pm 18$ mmHg because of the snout cone oxygen given together with inhaled anaesthesia. Considering that when an animal breathes only room air, the arterial pO2 is maximally at 104 mmHg [22, 23], the inhaled oxygen contributed a rise of 84 mmHg of pO2. As a similar amount of inhaled oxygen was administered to the ELO rats

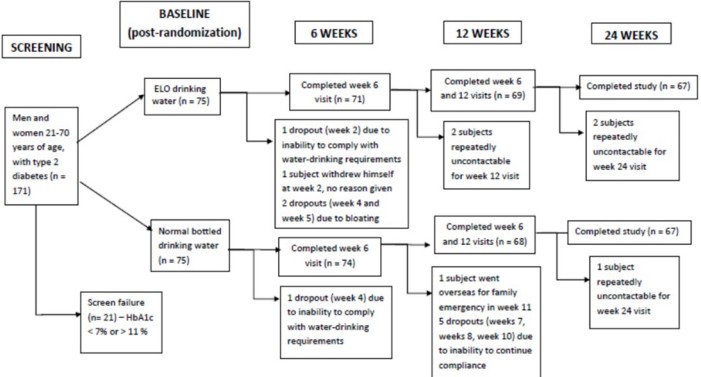

**Fig 1. Flow chart of study recruitment and dropouts.** Subjects had been advised that they would be able to withdraw from the study at any time. They were contacted at least a week before their next study appointment. Phone and/or SMS reminders were also given the day before their appointment.

during anaesthesia, inhaled oxygen would raise their arterial pO2 by a similar quantum (84 mmHg) to the control rats; therefore ELO water increased DO in arterial blood by 147 mmHg (335–188 mmHg). Improving the arterial plasma oxygenation by 147 mmHg of pO2 above the maximal baseline arterial pO2 of 104 mmHg normally contributed by pulmonary respiration constitutes a clinically significant oxygen supplementation from ELO water intake.

## Effects of ELO water vs. regular water on mitochondrial function in HepG2 cell cultures

In an environment with normal (5.5 mM) glucose concentration, the OCR linked to basal respiration was significantly higher (124.1 ± 16.9 pmol/min, 95% CI [107.3, 141.0], p = 0.002) in the ELO water-medium (220.6 ± 13.4 pmol/min) compared to the control (regular) water-medium (96.5 ± 9.3 pmol/min), The OCR linked to ATP production was also higher (94.0 ± 14.7 pmol/min, 95% CI [79.4, 108.7], p = 0.007) in ELO water (137.4 ± 12.9 vs. 43.4 ± 5.9 pmol/min, Fig 4). At high-glucose (25 mM) concentration, OCR linked to basal respiration was similar (29.3 ± 11.7 pmol/min, 95% CI [17.6, 41.1], p = 0.09) in ELO water

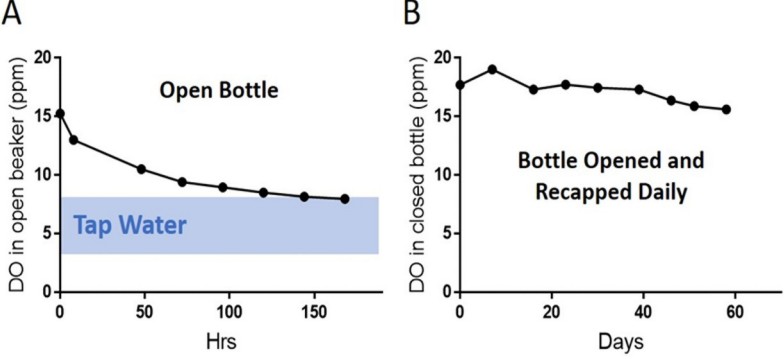

**Fig 2.** The dissolved oxygen (DO) level in ELO water measured using a dissolved-oxygen meter at sea level atmospheric pressure (A) in a bottle kept open for 7 days. The blue band indicates typical range of DO in tap water. (B) ELO water DO was measured in a bottle that was opened and recapped daily. DO was found to be still high at Day 58. The DO in ELO water is about 2 times higher than in tap water. This higher DO is stable, maintained over a considerable time period.

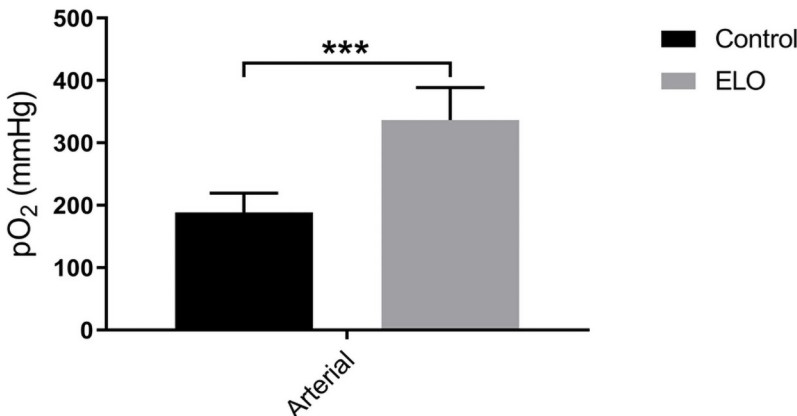

**Fig 3. Comparing arterial pO2 between ELO water-treated rats and rats drinking control water.** Arterial pO2 was significantly higher with ELO water (***, p < 0.01).

(230.0 ± 10.2 pmol/min) and control water (259.4 ± 5.0 pmol/min). In a high-glucose environment, there was no significant difference in OCR linked to ATP production (32.7 ± 16.6 pmol/min, 95% CI [16.2, 49.3], p = 0.09) in ELO or control water (170.0 ± 14.3 vs. 137.1 ± 7.0 pmol/min).

In the high-glucose environment, the OCR associated with proton leak across the mitochondrial membrane was significantly greater (62.1 ± 8.8 pmol/min, 95% CI [53.3, 70.8], p = 0.002) in ELO water (122.3 ± 7.3 pmol/min) compared to control water (60.3 ± 4.3 pmol/min). At normal glucose concentration, proton leak was also greater (OCR 30.1 ± 5.1, 95% CI [25.0, 35.2], p = 0.009) in cells incubated in the ELO water-medium (83.2 ± 2.1 pmol/min) compared to control water (OCR 53.1 ± 4.4 pmol/min).

## Effects of ELO water vs. regular water on mitochondrial mass and membrane potential

In cells cultured in the control (regular) water-medium, the high-glucose (25 mM) environment induced a reduction in mitochondrial mass (as measured with Mean Fluorescence

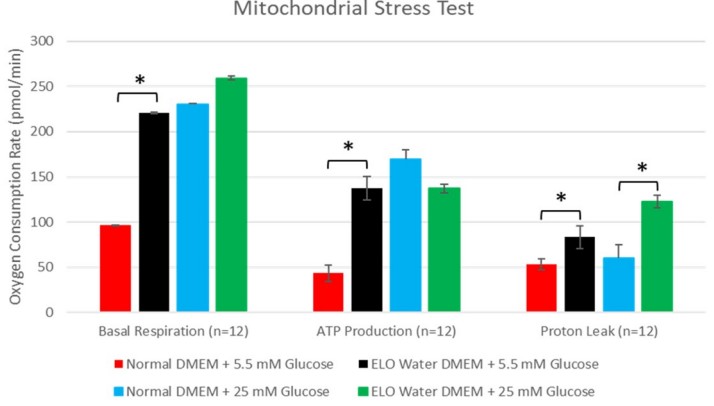

**Fig 4. Mitochondrial stress test: Oxygen consumption rates in HepG2 cells cultured in ELO water- and or control (regular) water-based media at normal (5.5 mM)- or high (25 mM)-glucose concentrations.** Significant differences between OCRs linked to basal respiration, ATP production and proton leak are indicated by * = p < 0.01.

Intensity MFI) by 1836 ± 735 arbitrary units (AU) when compared to a normal-glucose (5.5 mM) environment., In contrast, preservation of mitochondrial mass was demonstrated by an increase in MFI by 170 ± 642 AU in the ELO water medium cells in high-glucose environment compared to the normal-glucose environment (Fig 5A). This difference (MFI -2006 ± 325 AU, 95% CI [−2696, 1317]) was highly significant (p < 0.001). Similarly, the reduction in mitochondrial membrane potential (MP) induced with a high-glucose environment was more pronounced for cells cultured in the control water-medium (MFI -2256 ± 344 AU) compared with the ELO water-medium (MFI -614 ± 904 AU), resulting in a highly significant difference (MFI -1642 ± 322 AU, 95% CI [−2326, −959] p < 0.0001) shown graphically in Fig 5B.

### Effects of ELO water vs. standard bottled drinking water on glycaemic control in adults with type 2 diabetes

There were no significant differences in age, gender distribution, BMI or compliance with water (Table 1). There were no hypoglycaemic episodes requiring assistance, fluid retention, or electrolyte abnormalities. The dropout rate from both study arms was similar and largely due to inability to comply with water-drinking requirements (Fig 1). There were no significant differences in duration of diabetes or proportion of subjects on insulin treatment between the two groups (Table 1).

HbA1c was significantly lower at weeks 6 and 12 compared to baseline in the ELO water group, but did not change with placebo (Table 1). Compared to the placebo group, the reduction of HbA1c from baseline was significantly greater in the ELO group at 6 weeks (p = 0.03) after adjusting for baseline HbA1c. At week 6, 39.1% (25/71) of the ELO water group demonstrated reduction in HbA1c of at least 0.5% (5.5 mmol/mol) at 6 weeks, versus 18.9% (14/74) of the normal drinking water, which was statistically significant (p = 0.021). At week 12, ELO water was associated with significant (p = 0.002) decrease in HbA1c of 0.3 ± 0.7% (4 ± 8 mmol/L) from baseline. HbA1c did not change significantly in the placebo group during the study. Weight and fasting glucose did not change significantly in both groups.

## Discussion

We demonstrated that drinking ELO oxygen-enriched water was able to raise arterial oxygen levels. It is likely that the oxygen in ELO water was transported through the gut into the bloodstream through aquaporins which transport both water and oxygen [17]. We also found that mitochondrial respiration rates were elevated in cells cultured in ELO water as evidenced by higher oxygen-consumption rates associated with basal respiration and ATP-linked respiration, possibly facilitated by the oxygen-enriched environment. Mitochondrial dysfunction can be impacted due to cellular hypoxia [4]. Mitochondrial dysfunction is a central metabolic defect in the pathology of type 2 diabetes, as evidenced by reduced mitochondrial oxidative capacity [24], and lower mitochondrial respiration rates in diabetic individuals [25] compared with non-diabetic individuals which we also observed in our experiments. Pancreatic beta-cells are particularly prone to hypoxia mediated mitochondrial dysfunction because of the high oxygen requirement of insulin secretion [26]. During hyperglycaemia, the stimulus to upregulate insulin secretion therefore promotes hypoxia, and hypoxic effects on mitochondria have been found to increase production of mitochondrial reactive oxygen species (ROS) through heightened oxidative phosphorylation [27]. ROS can lead to inactivation of insulin gene transcription [28] and insulin secretion is further impaired by hypoxia-induced loss of the adaptive unfolded protein response (UPR) which normally prevents accumulation of misfolded or unfolded proteins (such as proinsulin) in the endoplasmic reticulum [29, 30], leading to a vicious circle of hyperglycaemia exacerbating cellular hypoxia through production of

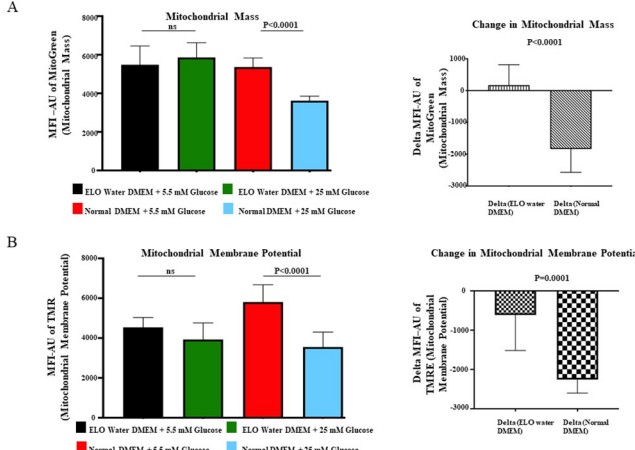

**Fig 5. Delta changes of Mean Florescence Intensity (MFI) in mitochondrial mass determined by MitoTracker Green (A) and mitochondrial membrane potential by TMRE labelling (B) in HepG2 cells cultured in ELO water-based DMEM or control (regular) water-based DMEM in glucose environments of normal (5.5 mM) or high (25 mM) glucose concentrations over 14 days.** The MFI, expressed in arbitrary units (AU), was used to represent the metric of MitoTracker and TMRE. Biological replicates (N = 3), Technical replicates (n = 3).

mitochondrial ROS [6]. ROS accumulation in the mitochondria, in a nutrient-rich environment of high glucose content similar to hyperglycaemia in uncontrolled diabetes, is mitigated by increasing proton leak across the mitochondrial membrane through uncoupling of ATP-synthase activity [31]. This mechanism in effect reduces the efficiency of oxidative phosphorylation in order to reduce ROS production and thus ROS-mediated damage to mitochondria. We observed greater increase in mitochondrial proton leak in the cells cultured with ELO water compared to normal-water controls when in a high-glucose environment, suggesting that ELO water may induce a protective response to hyperglycaemia-induced mitochondrial dysfunction, which if recapitulated in the pancreas could slow the progression of hypoxia-induced pancreatic beta-cell failure.

At high glucose concentrations, we demonstrated that mitochondrial mass and dysfunction (as measured by reduction in membrane potential) decreased in control-water media, consistent with hyperglycaemia causing fragmentation and apoptosis of mitochondria [32], and with the observations of Palmeira et al. [33] who reported that a high-glucose environment led to reductions in mitochondrial mass and mitochondrial membrane potential in HepG2 cells which they attributed to decreased mitochondrial biogenesis caused by progressive alteration in gene expression when exposed to a high-glucose environment. We additionally found that at high-glucose concentrations, incubation with ELO water preserved mitochondrial mass, and prevented the reduction in mitochondrial membrane potential, a sign of hyperglycaemia-induced mitochondrial dysfunction [34]. Taken together, our findings suggest that ELO water is protective against hyperglycaemia-induced mitochondrial dysfunction and attrition. A Japanese group observed that hyperglycaemia induced cellular hypoxia in diabetic mice and suppressed the expression of aquaporin-1 (AQP1, a water and oxygen channel which facilitates diffusion of oxygen across cell membranes) [6] while increased expression of AQP1 had a protective effect against hyperglycaemia-induced mitochondrial dysfunction, suggesting an association between alterations of transport of water and/or oxygen and mitochondrial dysfunction in high-glucose environments.

**Table 1. Baseline demographics, and changes in weight and glycaemic control from baseline, with ELO versus placebo drinking water.**

| | ELO water group (n = 75) | Change from baseline (Mean ± SD) | Placebo group (n = 75) | Change from baseline (Mean ± SD) | Difference between groups (Mean ± SEM) | P value (Difference between groups) |
|---|---|---|---|---|---|---|
| **Age** | 53.6 ± 9.4 | | 52.9 ± 8.7 | | | 0.594 |
| **Males** | 51 (68%) | | 51 (68%) | | | 1.000 |
| **BMI (kg/m$^2$)** | 28.5 ± 4.9 | | 27.7 ± 5.1 | | | 0.289 |
| **Insulin treatment** | 29 (38.6%) | | 23 (30.7%) | | | 0.298 |
| **Duration of diabetes (years)** | 12.3 ± 7.9 | | 12.3 ± 7.2 | | | 0.797 |
| **Volume of water (litres) drunk/week per subject** | | | | | | |
| Weeks 1 to 6 | 10.1 ± 0.9 | | 10.2 ± 0.5 | | -0.2 ± 0.1 | 0.208 |
| Weeks 7 to 12 | 10.1 ± 0.7 | | 10.2 ± 0.5 | | -0.1 ± 0.1 | 0.217 |
| **Weight (kg)** | | | | | | |
| Baseline | 81.4 ± 16.1 | | 78.5 ± 17.9 | | | |
| Week 6 | 81.6 ± 16.4 | 0.2 ± 1.1 | 78.6 ± 17.9 | 0.1 ± 1.3 | -0.2 ± 0.2 | 0.215 |
| Week 12 | 81.7 ± 16.2 | 0.3 ± 1.5 | 78.4 ± 18.0 | -0.1 ± 2.1 | -0.3 ± 0.3 | 0.345 |
| **HbA1c (%, mmol/mol)** | | | | | | |
| Baseline | 8.9 ± 1.1 (74 ± 12) | | 8.9 ± 1.0 (73 ± 11) | | | |
| Week 6 | 8.7 ± 1.2 (71 ± 13) | -0.2 ± 0.5 (3 ± 6)** | 8.8 ± 1.1 (73 ± 12) | -0.1 ± 0.6 (1 ± 7) | -0.2 ± 0.1 | 0.029 |
| Week 12 | 8.6 ± 1.2 (71 ± 14) | -0.3 ± 0.7 (4 ± 8)* | 8.8 ± 1.1 (73 ± 14) | -0.1 ± 0.9 (1 ± 10) | -0.2 ± 0.1 | 0.137 |
| **FPG (mmol/L)** | | | | | | |
| Baseline | 9.4 ± 2.5 | | 9.8 ± 3.0 | | | |
| Week 6 | 8.8 ± 2.6 | -0.6 ± 3.2 | 9.9 ± 3.2 | 0.1 ± 3.2 | -0.6 ± 0.5 | 0.305 |
| Week 12 | 9.3 ± 3.0 | -0.3 ± 3.4 | 9.9 ± 3.1 | 0.1 ± 2.8 | -0.1 ± 0.5 | 0.867 |

FPG = fasting plasma glucose. For changes from baseline,

*p < 0.01,

**p < 0.001.

ELO water improved glycaemic control within 12 weeks as an adjuvant to standard therapy for type 2 diabetes in our human subjects. The reduction in HbA1c of ~0.3% in the ELO water group occurred without hypoglycaemia or other adverse effects, and was comparable to the reduction in HbA1c by 0.29% with lifestyle-based weight loss interventions lasting at least 12 weeks in a meta-analysis of 5784 adults with type 2 diabetes [35]. Moreover, HbA1c decreased by at least 0.5% (5.5 mmol/mol) at 6 weeks, which is clinically significant [36], in ~40% of patients in the ELO water-group (approximately double the proportion in the placebo group). ELO water was thus associated with a significant improvement in glycaemic control, that was comparable to the reduction of mean HbA1c by 5.6 mmol/mol induced with medications over a similar period in the study of Hirst et al. [37]. The baseline average HbA1c of 8.7% (72 mmol/mol) in this study was similar to that of our population. Placebo water did not significantly improve HbA1c, suggesting that effects of ELO water are related to oxygenation rather than solely from improved hydration. Weight loss was not demonstrated in our study, so the improvements were unlikely to have been due to substitution of water for sugar-sweetened beverages, which was also been associated with lower blood glucose levels [13, 38, 39]. Reversal of hypoxia has been shown to improve insulin secretory capacity [40], insulin resistance [41], and beta-cell function [42]. In our study, ELO water reduced HbA1c significantly even in

individuals with diabetes for at least 10 years. Deterioration of beta-cell function which is thought to be irreversible after 10 years of diabetes [42], and decreased insulin secretion due to hypoxia-induced acute [43] and chronic [10] de-differentiation of beta-cells, contribute to worsening glycaemic control over time. β-cell failure and apoptosis have also been reported with hypoxia-induced loss of the adaptive UPR [29]. We thus hypothesize that the benefits of ELO water for glycaemic control are mediated by improving mitochondrial oxidative respiration, preserving mitochondrial mass and protection against mitochondrial dysfunction consequent to hyperglycaemia-induced cellular hypoxia which impairs insulin signalling [4, 30, 31].

The main limitations of this study are the short duration, the heterogeneous medication regime of our study population, and that there was no direct supervision (which would be challenging in this real-world study) of compliance with fluid intake, diet and physical activity. However, our subjects had been instructed not to alter their diet or physical activity patterns and their healthcare providers advised not to change their medications during the study period, while the majority of the participants reported compliance with water intake. Though we hypothesize that improved oxygen delivery may also be the basis of lowered glucose levels, we were not able to study the effects of ELO water on our human subjects' tissue oxygenation or mitochondrial function, nor measure ROS levels or markers of oxidative stress as possible mechanisms for the effects of ELO water. Another limitation of measurement was the lack of information on insulin resistance using insulin clamps, or surrogate measures such as HOMA-IR. Although our experiments were limited by not using adipocytes, myocytes or pancreatic cells to quantify insulin resistance, hepatic insulin resistance is a major contributor to hyperglycaemia in diabetes [44], and HepG2 cells have been used extensively in studies of mitochondrial function and in type 2 diabetes, NAFLD and other disorders of hepatic insulin resistance. We acknowledge that while Hep G2 cells are inherently different to primary hepatocytes, they are phenotypically stable, and more suitable for experiments of longer duration than primary hepatocytes, which are very difficult to obtain, have limited *in vitro* proliferative capacities and which quickly lose their hepatic phenotype and suffer poor viability in *ex vivo* environments [45]; moreover, HepG2 cells are a better model than fresh human hepatocytes to measure mitochondrial function [46]. Although we had considered a cross-over study with each subject as his/her own control, we hypothesized that benefits for glycaemic control would be mediated by metabolic reprogramming, such as improvement in mitochondrial function consequent to correction of cellular hypoxia would require an appropriate wash-out period in a cross-over study to have sufficient time to allow for reversal of metabolic alterations. As this is the first human diabetes study using oxygen-enriched water, an appropriate wash-out period for a cross-over study is still unknown, making a cross-over study design difficult to implement.

In conclusion, we found that ELO water raised arterial oxygen levels in animal models, improved cellular oxygenation and altered markers of mitochondrial function, and is an effective adjuvant therapy even in people with longstanding diabetes already on insulin. It would be useful to evaluate the effects of ELO water on mitochondrial function, oxidative stress and insulin resistance in a rat model of type 2 diabetes, as well as measuring insulin resistance and insulin secretion in patients consuming ELO water in combination with oral glucose-lowering medications and/or insulin.

## Supporting information

**S1 Checklist. Consort 2010 checklist.**
(DOC)

**S1 File. Study protocol.**
(DOCX)

## Acknowledgments

We would like to thank the participants of the study, Ms Chunfeng Tao, and Dr Mei-Yee Choy for her valuable insights.

## Author Contributions

**Conceptualization:** Joan Khoo, Christoph E. Hagemeyer, Eng-Kiong Teo.

**Data curation:** Joan Khoo, Christoph E. Hagemeyer, Darren C. Henstridge.

**Formal analysis:** Joan Khoo, Christoph E. Hagemeyer, Darren C. Henstridge, Sumukh Kumble, Ting-Yi Wang, Rong Xu, Carmen Kam.

**Funding acquisition:** Joan Khoo, Christoph E. Hagemeyer, Eng-Kiong Teo.

**Investigation:** Joan Khoo, Christoph E. Hagemeyer, Darren C. Henstridge, Sumukh Kumble, Ting-Yi Wang, Rong Xu, Linsey Gani, Thomas King, Shui-Boon Soh, Troy Puar, Vanessa Au, Eberta Tan, Tunn-Lin Tay.

**Methodology:** Joan Khoo, Christoph E. Hagemeyer, Darren C. Henstridge, Sumukh Kumble, Ting-Yi Wang.

**Project administration:** Joan Khoo.

**Resources:** Eng-Kiong Teo.

**Supervision:** Joan Khoo, Christoph E. Hagemeyer, Darren C. Henstridge.

**Validation:** Joan Khoo, Christoph E. Hagemeyer, Darren C. Henstridge.

**Writing – original draft:** Joan Khoo, Christoph E. Hagemeyer, Carmen Kam.

**Writing – review & editing:** Joan Khoo, Christoph E. Hagemeyer, Darren C. Henstridge, Sumukh Kumble, Ting-Yi Wang, Rong Xu, Linsey Gani, Thomas King, Shui-Boon Soh, Troy Puar, Vanessa Au, Eng-Kiong Teo.

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
