## [Decision Letter · Decision Letter 0]

16 Oct 2020

PONE-D-20-22349

Effects of water stably-enriched with oxygen as a novel method of tissue oxygenation on mitochondrial function, and as adjuvant therapy for type 2 diabetes in a randomized placebo-controlled trial

PLOS ONE

Dear Dr. Khoo,

Thank you for submitting your manuscript to PLOS ONE. After careful consideration, we feel that it has merit but does not fully meet PLOS ONE’s publication criteria as it currently stands. Therefore, we invite you to submit a revised version of the manuscript that addresses the points raised during the review process.

Please address each of the reviewers comments.

We look forward to receiving your revised manuscript.

Kind regards,

Gordon Fisher

Academic Editor

PLOS ONE

Journal Requirements:

"The study was sponsored by Elomart Pte Ltd, Singapore".

We note that you received funding from a commercial source: 'Elomart Pte Ltd'.

Reviewers' comments:

Reviewer's Responses to Questions

**Comments to the Author**

1. Is the manuscript technically sound, and do the data support the conclusions?

Reviewer #1: No

Reviewer #2: No

2. Has the statistical analysis been performed appropriately and rigorously? 

Reviewer #1: No

Reviewer #2: Yes

3. Have the authors made all data underlying the findings in their manuscript fully available?

Reviewer #1: No

Reviewer #2: Yes

4. Is the manuscript presented in an intelligible fashion and written in standard English?

Reviewer #1: Yes

Reviewer #2: Yes

5. Review Comments to the Author

Reviewer #1: This paper looks at a study in rats and a randomised controlled trial of ELO vs tap water in individuals with diabetes. At present the design characteristics of the randomised trial are not given - why was 150 participatns chosen; how was randomisation performed etc.

Please give exact p-values throughout and where t-tests etc have been performed give the difference and 95% CI. The dynamite plots currently give less information than a table with the actual data. Numbers are sufficiently small to give actual data points anyway - so the dynamite plots obfuscate the data.

The analyses in Table 1 are wrong. It is entirely incorrect to analyse within group - the between group differences and the p-values should be given and reported. Given the dropouts how many patients were analysed at each follow-up timepoint - was there any imputation of missing data?

Please remove the subgroup analysis unless a proper test for interaction demonstrates heterogeneity.

Reviewer #2: Khoo and colleagues have provided a nicely-written manuscript describing their study examining the effect of oxygen-enriched bottled drinking water (ELO) on metabolic endpoints in HepG2 cells, and also the results of a pilot study of the effects of ELO on HbA1c and other metabolic parameters in a cohort of T2 diabetes patients. Improvements in arterial blood oxygenation have, in some previous reports, provided evidence of improvements in metabolic efficiency and glycemic control, so there is some justification for using ELO in this manner. There are a number of major limitations to the study in its current form which substantially diminish its impact and significance to the field, however. Some of these are described below.

1) The cell model chosen for the in vitro study is not appropriate. HepG2 cells are immortalized and metabolically are not at all similar to primary hepatocytes, and great care must be taken with these cells to make sure they are phenotypically similar to primary heps. There is no mention of these steps at all by the authors, and in any case, the rationale for using hepatocytes is weak, since these cells do not make sense to study in the context of T2 diabetes. Authors should choose consider using myocytes, adipocytes, or even pancreatic beta cells as a model, to determine if ELO has an impact on the metabolism of these cells. And this leads to the next comment,

2) The increases seen in mitochondrial O2 concentration do not seem to be exclusive to ‘proton leak’, but rather all across the board, and this strongly suggests that ELO has simply increased mitochondrial content in these cells, which is not particularly surprising, but somewhat interesting. Authors need to provide evidence for changes in mitochondrial content in these cells.

3) If the authors wish to make the argument that ELO alters redox environment or changes levels of oxidative stress in the body, or in the cells, then they need to provide evidence of this in their study. No markers or readouts of ROS or antioxidants were performed here at all.

4) Why did the authors not examine the effect of ELO more thoroughly in the rat model, specifically in a rat model of T2 diabetes? There are many options available, genetic- or diet-induced obese rat models, that are widely used. This would allow for a much more thorough interrogation of the effect of ELO.

5) The drop in HbA1c seen in the T2 diabetes patients with ELO water is somewhat interesting, but would be hard to argue that a drop from 8.9 to 8.6 is clinically meaningful.

6) It is disappointing that the authors did not measure fasting serum insulin levels in their pilot study, as this would have allowed them to determine if ELO has altered insulin sensitivity in the patients through the HOMA-IR index.

6. PLOS authors have the option to publish the peer review history of their article (what does this mean?). If published, this will include your full peer review and any attached files.

Reviewer #1: No

Reviewer #2: No

---

## [Author Response · Author response to Decision Letter 0]

11 Apr 2021

Please see the uploaded file of Response to Reviewers.

---

## [Decision Letter · Decision Letter 1]

5 May 2021

PONE-D-20-22349R1

Effects of water stably-enriched with oxygen as a novel method of tissue oxygenation on mitochondrial function, and as adjuvant therapy for type 2 diabetes in a randomized placebo-controlled trial

PLOS ONE

Dear Dr. Khoo,

Thank you for submitting your manuscript to PLOS ONE. After careful consideration, we feel that it has merit but does not fully meet PLOS ONE’s publication criteria as it currently stands. Therefore, we invite you to submit a revised version of the manuscript that addresses the points raised during the review process.

Please address the comments by Reviewer 1

We look forward to receiving your revised manuscript.

Kind regards,

Gordon Fisher

Academic Editor

PLOS ONE

Journal Requirements:

Reviewers' comments:

Reviewer's Responses to Questions

**Comments to the Author**

1. If the authors have adequately addressed your comments raised in a previous round of review and you feel that this manuscript is now acceptable for publication, you may indicate that here to bypass the “Comments to the Author” section, enter your conflict of interest statement in the “Confidential to Editor” section, and submit your "Accept" recommendation.

Reviewer #1: (No Response)

Reviewer #2: All comments have been addressed

2. Is the manuscript technically sound, and do the data support the conclusions?

Reviewer #1: Partly

Reviewer #2: Yes

3. Has the statistical analysis been performed appropriately and rigorously? 

Reviewer #1: Yes

Reviewer #2: Yes

4. Have the authors made all data underlying the findings in their manuscript fully available?

Reviewer #1: Yes

Reviewer #2: Yes

5. Is the manuscript presented in an intelligible fashion and written in standard English?

Reviewer #1: Yes

Reviewer #2: Yes

6. Review Comments to the Author

Reviewer #1: Thank you for your responses to my previous comments. I have a few areas where the responses are not clear:

I understand the approach to build the trial based upon a standardised effect size - but such effect sizes are not necessarily the clinically relevant difference which is on the absolute scale. What absolute difference does d=0.5 correspond to, and why is this an appropriate choice?

It is well known (see the papers dating from the mid-1990s onwards) that envelope randomisation is prone to issues with allocation foreknowledge (indeed the classic case given by Altman in his textbook is an envelope randomisation). What steps were taken to ensure that envelopes were opened in sequence only after enrollment?

LOCF can be anti-conservative under certain circumstances - are results robust to method of imputing missing data?

Reviewer #2: The additional mitochondrial content measurements are informative about the potential for ELO to rescue mitochondrial content loss with hyperglycemia.

7. PLOS authors have the option to publish the peer review history of their article (what does this mean?). If published, this will include your full peer review and any attached files.

Reviewer #1: No

Reviewer #2: **Yes: **Ethan Anderson

---

## [Author Response · Author response to Decision Letter 1]

2 Jun 2021

Please see attached cover and reviewer reply letters.

---

## [Editor Report · Decision Letter 2]

1 Jul 2021

Effects of water stably-enriched with oxygen as a novel method of tissue oxygenation on mitochondrial function, and as adjuvant therapy for type 2 diabetes in a randomized placebo-controlled trial

PONE-D-20-22349R2

Dear Dr. Khoo,

We’re pleased to inform you that your manuscript has been judged scientifically suitable for publication and will be formally accepted for publication once it meets all outstanding technical requirements.

Kind regards,

Gordon Fisher

Academic Editor

PLOS ONE
---

## [Editor Report · Acceptance letter]

5 Jul 2021

PONE-D-20-22349R2 

Effects of water stably-enriched with oxygen as a novel method of tissue oxygenation on mitochondrial function, and as adjuvant therapy for type 2 diabetes in a randomized placebo-controlled trial 

Dear Dr. Khoo:

I'm pleased to inform you that your manuscript has been deemed suitable for publication in PLOS ONE. Congratulations! Your manuscript is now with our production department. 

Kind regards, 

on behalf of

Dr. Gordon Fisher 

Academic Editor

PLOS ONE